# The Flow of Institutional Charisma: Quanzhen Taoism and Local Performing Arts in Republic Shandong and Henan

**Guoshuai Qin** [1,*] **and Wanrong Zhang** [2]

[1] Institute of Culture and Industry, Qilu University of Technology, Ji'nan 250353, China
[2] School of Philosophy, Fudan University, Shanghai 200433, China
[*] Correspondence: qinguoshuai@qlu.edu.cn

**Abstract:** Quanzhen Taoism and its relationship with local performing arts is an important yet inadequately studied subject, to date, due to the shortage of and limited access to new sources. However, on the basis of historical documents, oral statements and field research, we determined at least eight genres of local performing arts closely related to Quanzhen Taoism, especially its sublineage, the Longmen School, in Republic Shandong and Henan. They traced back their own history to Quanzhen Taoist patriarch WANG Chongyang, adopted the Quanzhen Taoist lineage poem to name their disciples, and created the Ever Spring Guild (Changchun hui 長春會), in the name of Quanzhen Taoist QIU Changchun 丘長春, to assist each other. In other words, the Quanzhen Taoist institution was imitated by the local performing arts and, at the same time, the local artists performed some reasonable adaptations and accommodations to meet their own needs. By reviewing the local performing arts in Shangdong and Henan provinces, we can further understand Quanzhen Taoism in popular cultural traditions and local societies.

**Keywords:** Quanzhen Taoism; institutional charisma; local performing arts

## 1. Introduction

Ever since its beginning, Quanzhen Taoism has been consciously established as an institutionalized order, characterized by its ever-lasting emphasis on inner alchemy, original genealogical naming system, Chan-like monastic rules and distinctive religious practices, such as begging and cloud-travel (*yunyou* 雲遊) (Goossaert 1997). Additionally, not surprisingly, we can identify the elements and influences of Quanzhen Taoism in various avenues in society (Goossaert 2007; Katz 2000). On the issue of its relationship with the performing arts (see Durkheim 1971; Strickmann 2002; Van der Loon 1977; Chen 2007), modern scholars have convincingly demonstrated that most of the popular theatrical genres in 13th and 14th centuries can easily determine their origins in the religious lore of the Quanzhen Taoism of the period (Hawkes 1981; H. Wang 2004; Zuo 2004). In addition, Quanzhen Taoism can also be expressed in poetry, including lyrics (*ci* 詞), songs (*ge* 歌), verses (*lüshi* 律詩), and quatrains (*jueju* 絕句) (Komjathy 2013), even to the extent that "no any other religion [in the Yuan dynasty] could draw more attention from the literati community or exert more influence upon the literature world than Quanzhen Taoism" (Deng 1991, p. 23).

We should note that all the abovementioned scholars' arguments are grounded in the textual evidence and canonical sources that focus on the influence of Quanzhen teachings and masters upon the themes of theatrical genres. However, these studies barely examine the impact of the Quanzhen Taoist institution on the organization of local artists and their performing troupes. Additionally, the tight fictive kinship signified by the lineage poem (*paishi* 派詩) and the legendary patriarchs QIU Chuji 丘處機 (Taoist name: QIU Changchun 丘長春, 1148–1227) and his enduring appeal, are all typical elements of Quanzhen Taoism that appear in the practice of performing local arts.

These undeveloped aspects, we venture to say, have inspired our conception of "institutional charisma", the phrase we used in the title of this paper. As perceived by Quanzhen

scholars, Quanzhen Taoism, starting from QIU Chuji's journey westward to meet Genghis Khan in the Hindu Kush, gradually evolved into an institutional order with a national distribution (Kubo 1967; Yao 1980a; Goossaert 1997; Mou et al. 2005; Komjathy 2007). In its institutionalization, of course, Quanzhen Taoism formed a set of concepts to create its own identity, of which the genealogical system was at the core. Although having almost nothing in common with the lineage poem discussed later in this paper, the naming system utilized by Quanzhen Taoists in the Jürchen-Jin and Mongol Yuan dynasties, namely, ZHI 志, DAO 道 and DE 德 for the males, and MIAO 妙, SHOU 守 and HUI 慧 for the females, was an obvious indicator for identifying a Quanzhen cleric (Goossaert 2001, pp. 131–32). Similar to the naming system, sacrifices performed for Quanzhen patriarchs was an annual event for Quanzhen Taoists, which was a formal occasion used to meet the same practitioners from different parts of the country and share their mutual views and collective identity created by the same patriarch. For example, when QIU Chuji was interred at the White Cloud monastery in 1228, over 10,000 Quanzhen Taoists from all areas of the country gathered to attend the funeral rite (W. Zhao 2005, p. 417), and surely it was the first time that most of them, and possibly their disciples, had met each other face to face and knew of their existence per se. Another less-mentioned, but equally crucial, concept for the institutionalization of Quanzhen Taoists was the construction of its sacred history. We know for sure that WANG Chongyang was the founder of Quanzhen Taoism; however, according to QIN Zhian 秦志安, a Quanzhen Taoist of the third generation, the legendary, immortal donghu dijun 東華帝君was the first Quanzhen partiarch, and consequently, Quanzhen Taoist history would be as long as, if not longer than, that of Zhenyi Taosim 正一道 (Goossaert 2021). In their competition with Zhengyi Taoism, Quanzhen Taoists included this attributed history into the Taoist canon *Xuandu baozang* 玄都寶藏 as the symbolic capital to prove their antiquity and orthodoxy (Chia 2011, p. 169; Komjathy 2013, p. 6). While reviewing the sources with regard to the local performing arts in the provinces of Shandong and Henan, we found that they made frequent references to Quanzhen Taoism in many places and from time to time, and mostly focused on its lineage poem, patriarchal sacrifice and history, all of which constitute the institutional form of the Quanzhen order, other than the cultivating practices, such as inner alchemy, asceticism and so forth. In other words, the institutional organization is the charismatic legacy left by Quanzhen Taoism and inherited by local artists.

The local performing art (*difang quyi* 地方曲藝) include singing and narrations, where performers use a third-person voice to narrate in the vernacular, possibly accompanied by simple musical instruments. The number of performers is usually one or in pairs, but does not exceed five individuals. This form of performance emerged in the Tang Dynasty (618–907) and became popular because of its accessibility and colloquialism, and there are probably thousands of different types of *difang quyi* that have passed around China throughout history. As previously mentioned, the field data and oral accounts from the Republican times (1912–1949) convey an interesting image of the relationship between Quanzhen Taoism and the organizational structure of the local performing arts: in the Shandong 山東 and Henan 河南 provinces, Quanzhen masters served as patriarchs and patrons gods, different branches in the lineages of local performers distinguished themselves from each other on the basis of the lineage poems used by different Quanzhen lineages and they established the theatrical Ever Spring Guild (Changchun Hui 長春會) under the name of Quanzhen master QIU Changchun. With reference to Rolf Stein's "ceaseless dialectical movement of coming and going" between Taoism and local cults, we can venture to say that Quanzhen Taoism predominated its dialectical movements with the local performing arts by virtue of its own institutional charisma.

First, we must provide a brief clarification of the sources. Our arguments mainly focus on the Republican era, and the period beyond that time, when necessary; therefore, in addition to the social investigations and statistics provided at present, we also adopt part of the oral accounts as our source. These oral accounts are not the products of our own fieldwork, but were collected by teams of scholars performing systematic surveys at that time.

The most important text is the *Investigation of Popular Entertainment in the Xiangguo Temple* (Xiangguoshi minzhong yule diaocha 相國寺民眾娛樂調查, 1936, reprint in 1989), which is a survey of local folklore and entertainment for the purpose of implementing a new form of popular education. The investigators expressed a sympathetic critical reflection of the performers. Since the mid-nineteenth century, Xiangguo temple has become a local center for social activities of Kaifeng 開封 in Henan, with various local actors performing in or around the temple (Bianweihui 1995, pp. 459–61).[1] Our other major source is the *Henan quyi zhishi ziliao huibian* 河南曲藝志史資料彙編 (Bianjibu 1988, 1989, 1990, hereafter *QYHB*), which was edited in the 1980s and presents over a hundred articles on various 20th-century performing arts in Henan province. It is part of a larger national survey project of the performing arts, directed by the Ministry of Culture and other government departments and conducted by local scholars and cultural workers, and was intended to provide a systematic understanding of the surviving performing arts in China. The articles in the *QYHB* are mostly surveys conducted at the time or excerpts from other sources, and a few of them provide obscure or limited information of the survey. To better understand the source, we also interviewed Xu Liping 徐立平, a contemporary popular performer of the Shandong Laozi 山東落子 in Ji'nan 济南, who specializes in more than forty pieces and has greatly contributed to the transmission of this performing art. The sources we have at our disposal, to date, mainly concern Shandong and Henan in the Republican period; therefore, we use the performing arts in these two provinces as the subject of our discussion. The reliability and accuracy of the oral accounts are by no means unquestionable,[2] especially concerning the origin and history of certain local performing art genres, in which the performers express their specific understanding of their relationship with Quanzhen Taoism. Moreover, we should remain prudent when using these materials to develop our discussion.

Whatever the origins and histories, though quite fascinating, of these dramatic traditions, we are still interested in their relations with Quanzhen Taoism, in particular, and thus, we exclusively address their formal organizations that are borrowed directly from the Quanzhen Taoist institution. How did local performing arts use the lineage poem of the Longmen School for the master–disciple transmission? What influence did the Quanzhen institution have on sacrificial ceremonies, disciple initiation rites and organizational forms of local performing troupes? How did they modify Quanzhen history, the legends of the patriarch QIU and the lineage poem? By addressing these questions, we progressively reveal the distinctive influence of the institutional tradition of Quanzhen Taoism on the local performing arts in Shandong and Henan in the Republican period.

## 2. Local Performing Arts and Their Internal Succession

According to the extant sources, there are at least eight local genres of performing arts in the provinces of Shandong and Henan. We were not able to describe in detail the types of performance, instruments, repertoire and the other details of each genre; however, we presented their basic characteristics. In order to keep to the theme of this study, we directly address the point that the transition of these local performing arts from master to disciple is based on the lineage poem of the Longmen School. Additionally, when relating to their origins, they all worship the Quanzhen master QIU Changchun as their patriarch and create their own sacred history by following the Taoist pantheon of immortals. However, although these eight performing arts share the same Longmen lineage poem, each genre is divided further into various branches in their transmission to different regions; each branch is named after their different founding master so as to distinguish themselves from each other, such as Dongzhang men 東張門 and Xizhang men 西張門.[3]

### 2.1. Taoist Ballade

The Taoist ballade (Daoqing 道情) is a narrative art closely related to Taoism. It was initially performed by Taoists to propagate Taoism and eulogize Taoist teachings, accompanied by drums (*yugu* 漁鼓[4]). The themes were mainly based on Taoist thoughts and stories of the gods and immortals. It probably arose during the Tang and Song (960–1279) dynas-

ties and gradually developed into a kind of art performed by professional local actors after the Yuan (1271–1368) and Ming (1368–1644) dynasties. Its themes also went beyond Taoist stories and were widely spread in several northern and southern provinces.[5]

In Republican times, this genre was largely performed by laymen who worshipped none other than QIU Changchun as their patriarch (Ma 1989, p. 47). A Republican-era sociological investigation indicated that there were still five people in 1936 chanting the Taoist ballade in the Xiangguo temple of Kaifeng, Henan: ZHANG Yuancai 張元材 and his disciple QIAO Mingsong 喬明松, MA Lichun (courtesy name: Hongbin) 馬禮純 (宏斌) and his two disciples YANG Zongshan 楊宗山 and ZHANG Zongxi 張宗西. Among these five persons, MA Lichun was the leader with the most power and greatly influenced Daoqing and other performers. He also had four apprentices: YANG Zongyi 楊宗義, ZHAO Zongnan 趙宗南, ZHAO Zongbei 趙宗配 (北) and XU Zongdong 徐宗洞 (東) (L. Zhang 1936, p. 130). From these two pairs of master–disciple relations, we could conclude that the Taoist ballade performers in Xiangguo temple were affiliated with the Longmen School, and that they, respectively, belonged to four generations, identified by the names of Yuan, Ming, Li and Zong.

Moreover, the deceased Taoist ballade performer DONG Mingde 董明德 from Taikang County 太康縣 in Henan province possessed a copy of the *Hundred-Character Chart for the Taoist Name and Genealogy of the performers from the Longmen School of Patriarch QIU* (Qiuzu longmenpai yiren daohao beixi baizipu 邱祖龍門派藝人道號輩系百字譜) (QYHB, 3, pp. 226–27), whose generation order was the same as the of the *General Register of Lineages Revealed by Various Veritables* (Zhuzhen zongpai zongbu 諸真宗派總簿), preserved, at present, in the White Cloud Monastery 白雲觀 in Peking.

### 2.2. Henan Zhuizi

Henan Zhuizi (墜子 or 墜字) is generally considered to be a derivative of the Taoist ballade, while synthesizing other performing arts formed around the end of the Qing Dynasty, popular in the Henan region and spreading to Tianjin 天津 and other places. Its appellation comes from the fact that the performers use a two-stringed zither (*qin* 琴) that creates a falling sound or because they drop the tone of the last word of each sentence, and the performers are usually one or two people (Zhang 1951; *QYHB*, 2, pp. 108–26).

Some of the earliest Zhuizi performers were Taoist balladists. For example, LEI Ming 雷明, who performed in Kaifeng in 1905, continued to name his disciples based on his lineage poem after he began to perform Zhuizi (*QYHB*, 3, p. 13). ZHANG Mingliang 張明亮, who performed at Xiangguo temple in the 1930s, is a similar example. He started learning and performing Taoist ballades at twelve years old and had five disciples of the "Zhi 至" generation. At the age of thirty-five years, he felt that he could not maintain his daily life by performing Taoist ballades; therefore, he decided to perform Zhuizi and then had ten disciples, two males and eight females, from the same "Zhi" generation. After 1921, more and more women participated in Zhuizi performances, and they also used the lineage poem of the Longmen School (L. Zhang 1936, p. 93).

In Republican Henan, there were 36 performers from seven tea gardens (*chayuan* 茶園, the name of the troupe) who performed Zhuizi at Xiangguo temple, of which FAN Lifeng 范禮鳳 and ZHANG Licui 張禮翠 from the Qingchun Tea Garden 青春茶園 were the most outstanding (L. Zhang 1936, pp. 78–79). Among the 36 performers, 22 belonged to the Longmen School, corresponding to six generations: Jiao 教, Yong 永, Yuan 元, Ming 明, Zhi 芝/ 治 and Li 禮/ 理. Moreover, their succession from master to disciple was strictly in accordance with the lineage poem of the Longmen School.

In addition, according to the theatrical lineage manuscript held by ZHANG Mingliang, the Zhuizi genre can be divided into "Seven Veritables and Eight Lineages" 七真八派: the "Seven Veritables" were no other than the "Seven Veritables of Quanzhen Taoism" 全真七子, but with a slight variance, namely, QIU Chuji 丘處機, LIU Langyan 劉朗言, TAN Changsheng 譚長生, MA Danyang 馬丹陽 (MA Yu, 馬鈺, 1123–1183), HAO Taigu 郝太古 (HAO Datong 郝大通, 1140–1212), WANG Yuyang 王玉陽 (WANG Chuyi 王處一,

1142–1217) and SUN Qingjing 孫清靜 (SUN Bu'er 孫不二, 1119–1182); as for the "Eight Lineages", except for the seven lineages of Longmen 龍門, Suishan 隨山, Nanwu 南無, Yushan 遇山, Huashan 華山, Lunshan 侖山 and Qingjing 清靜, founded by the Seven Veritables, respectively, there was an eighth lineage called Yinxi 寅戲 (L. Zhang 1936, pp. 74–77). Compared with the lineage poem of the *General Register of Lineages Revealed by Various Veritables*, there was no difference in the order of these characters, except for some generation characters incorrectly written, but pronounced the same. Among them, LIU Langyan should be LIU Changsheng 劉長生 (original name: LIU Chuxuan 劉處玄, 1147–1203) and TAN Changsheng should be TAN Changzhen 譚長真 (original name: TAN Chuduan 譚處端, 1123–1185). As for the eighth school, "The tiger elder brother (Laohu dashixiong 老虎大師兄) created the Yinxi school, which was the eighth school, and this school had no successor" (L. Zhang 1936, p. 77). Possibly, this theatrical Yinxi school corresponds to the Taoist lineage Yinxi pai 尹喜派, whose lineage poem is preserved in the *General Register of Lineages Revealed by Various Veritables*; however, unfortunately, no further information has been discovered.

Moreover, in the 1940s, the performer ZHANG Yuanfa 張元法 and his two disciples, LI Mingliang 李明亮 and ZHANG Mingyue 張明月, performed the "disguised Zhuizi " 化裝墜子 in Neihuang County 內黃縣, Henan (*QYHB*, 3, p. 63). According to the middle character of their given names and their relationship as master and disciple, they also seemed to belong to the Longmen School.

### 2.3. *Shuoshu*

Storytelling (Shuoshu 說書, also called Pingci 評詞) is a form of language performance that originated in the Tang and Song (960–1279) dynasties, with a single performer who generally only uses simple gestures, does not sing and does not use musical instruments. The stories they tell, mainly historical, have rich themes, some of which have a strong Taoist perspective (*QYHB*,1, p. 62).

In the Republican era, there were eight story-tellers still active in Xiangguo temple: DAI Mingyin 戴明印, CHU Zhigang 楚至鋼, WANG Futang 王福堂, FAN Mingxian 范明顯, ZHU Yuanhui 朱元慧, ZHOU Mingyuan 周明元, WANG Mingshun 王明順 and WU Mingwen 武明文 (L. Zhang 1936, p. 103). Except for the unidentified WANG Futang, the other seven performers belonged to the Longmen School. It is worth noting that, among these Pingci Shuoshu performers, there was a direct or indirect master–disciple or coreligionist relation between them, all descending from a story-teller named LI Yongxue 李永學, as showed in the following Table 1.

**Table 1.** The lineage of LI Yongxue (L. Zhang 1936, pp. 63–74).

| | | | | |
|---|---|---|---|---|
| LI Yongxue | MA Junting | WANG Wanfu | | |
| | | WAGN Mingshun | | |
| | | LI Mingfu | ZHANG Zhizhong | WANG Ligui |
| | | MA Mingtang | CHU Zhigang | |
| | | QIU Dacheng | | |
| | DUAN Yuanshan | FAN Mingxian | | |
| | | ZHOU Mingyuan | | |
| | | LIU Mingxiang | | |
| | | DAI Mingyin | FENG Zhiying | |

Except for this performing group in Xiangguo temple, there were two other famous *Pingci Shuoshu* performers, ZHAO Yuancheng 趙元城 and his disciple JI Mingjun 紀明君, whose successive transition from master to disciple was as follows (Table 2):

**Table 2.** The lineages of ZHAO Yuancheng and JI Mingjun (L. Zhang 1936, pp. 63–74).

| | | |
|---|---|---|
| YUAN Yongtang | ZHAO Yuancheng | ↘ |
| WEI Yonghai | ZHANG Yuande | JI Mingjun |

What is notable is that, in addition to the common rite of worshiping QIU Changchun as the patriarch, there was also a saying of "Seven Veritables and Eight Lineages" circulating among the Pingci Shuoshu performers, which was the same as that of Henan Zhuizi, except for some individual generation characters (*QYHB*, 1, p. 60).

### 2.4. Shandong Laozi

Shandong Laozi 山東落子 is a sub-lineage of the Lotus Rhyme (Lianhua Lao 蓮花落). The Lotus Rhyme is performed during begging, or possibly by Buddhist monks during begging and fundraising activities and has been popular since the Song Dynasty. In the Jiaqing 嘉慶 period (1760–1820) during the Qing Dynasty (1644–1911), it was created in Shandong by combining local dialects and folk songs, mostly performed by a single person or a pair, using a large cymbal and a bamboo board as instruments.

In the Republican era, the most well-known Shandong Laozi performer was GU Hezhen 顧合真 who belonged to the Longmen School and was mostly active in the Fan county 範縣 of Henan province and the rural areas of Heze 菏澤 and Liaocheng 聊城 in Shandong province. According to a study, at that time, most of the Shandong Laozi performers belonged to ZENG 曾, CHAI 柴, YAN 閻 and ZHANG 張, the four dominating theatre branches of the Longmen School, and some other performers belonged to the branches of Sunzhao 孫趙, Meiqing 梅清 and Dingsi 丁四. They all worshiped Quanzhen master QIU Changchun as their patriarch, claimed themselves to be the successors of the Longmen School and set up a one-hundred-character genealogy; the active performers at that time corresponded to the eighteen to twenty-three-generation characters (*QYHB*, 2, pp. 155–56).

It is worth mentioning that, at present, XU Liping is still performing Shandong Laozi in the Changqing 長清 district, Ji'nan 濟南 City, Shandong province (Figure 1). To date, he is one of the most famous performers in the area, and he proves that such a tradition still exists. XU Liping, using XU Yongkui 徐永奎 as his stage name, acknowledge performer QU Jiaowen 曲教文, who is the disciple of CHEN Heyun 陳和雲, as his master in the early 1950s, and learned Shandong Laozi from him. Xu Liping states that he belongs to the Dongzhang branch 東張門 of the Longmen School.[6] Additionally, he also said that, although he knew he belonged to the Longmen School and that his patriarch was QIU Chuji, he did not know why.

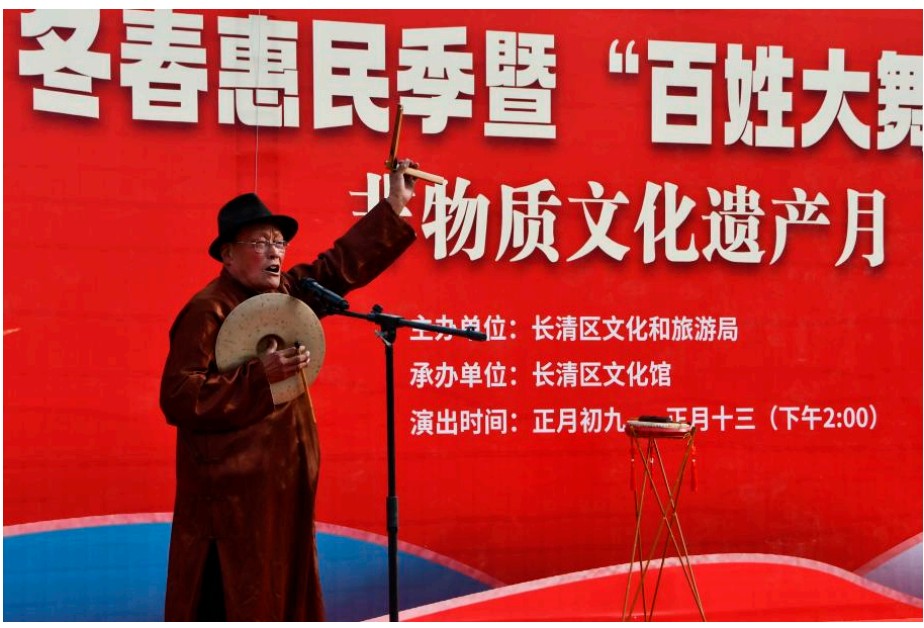

**Figure 1.** XU Liping performing the Shandong Laozi at the Changqing culture fair on 31 January 2023.

### 2.5. Shandong Kuaishu

Kuaishu 快書 existed during the mid-Qing Dynasty era and was popular in Shandong province. It is famous for telling the story of WU Song 武松 in *Water Margin* (Shuihu zhuan 水滸傳) and can be performed by one or two people using musical instruments, such as the three-stringed zither (*sanxian* 三弦) (X. Guo 2004). These performers worshiped Quanzhen Taoist QIU Changchun as their patriarch. That is to say, as far as its affiliation is concerned, Shandong Kuaishu also belonged to the Longmen School. From the Republican era to the present day, the relatively well-known Shandong Kuaishu performers were QI Yongli 戚永立 and his disciple GAO Yuanjun 高元钧, who both belonged to the Laozhang branch 老張門, mostly active in Shandong province (Jiang 2009, p. 126; Li 2011).

### 2.6. Guangzhou Dagu

Guangzhou Dagu 光州大鼓 are performed by one person. The performer sings alongside drums as an accompaniment and plays multiple roles. It appeared during the Xianfeng 咸豐 period (1851–1861) during the Qing Dynasty, and was mainly popular in southeast Henan (*QYHB*, 2, p. 158). At that period, the performers in the Huangchuan 潢川 area of Henan were CHANG Hebin 常和賓, CHANG Jiaozhi 常教芝, WEI Yuanzong 魏元宗, LIU Yuanzhong 劉元中 and LIU Yuanpeng 劉元鵬 (Bianweihui 1996, p. 1662). They belonged to the He 和, Jiao 教 and Yuan 元 generations, while performers of the Yong 永 generation, in between the Jiao and Yuan generations, are missing.

According to the oral accounts, there is a popular saying among the troupe performing the *Guqu* 鼓曲 genre of "Seven Veritables, Eight Lineages and One Hundred Branches": Quanzhen master and patriarch Wang Chongyang propagated his teaching by performing *Guqu*, and extensively adopted and taught his disciples. He adopted one hundred disciples in total, with seven of them "attaining the Tao": QIU 邱, LIU 劉, TAN 譚, MA 馬, HAO 郝, WANG 王 and SUN 孫, namely, the "Seven Veritables"; later on, QIU Changchun inherited his master's teaching and created the Longmen School; QIU has eight successful disciples whose surnames are GAO 高, GUI 桂, CHAI 柴, ZHANG 張, XING 省, ZHAO 趙, HAN 韓 and YANG 楊, namely, the "Eight Lineages"; these eight people continued to recruit and teach their disciples; therefore, *Guqu* was passed on from generation to generation. Although the term "Seven Veritables, Eight Lineages" is the same as in the Zhuizi, the "Eight Lineages" refer to different theatrical branches, not the schools of Quanzhen Taoism. The Guangzhou Dagu performers also agreed with this saying and left behind the *Hundred-Character Chart for the succession of Longmen School* (Longmenpai baizi chuancheng zipu 龍門派百字傳承字譜) (*QYHB*, 3, pp. 38–40).

### 2.7. Puyang Qinshu

The appellation Qinshu 琴書 comes from the accompanying instrument, the yangqin 扬琴, which was formed during the Qianlong 乾隆 period (1736–1796) during the Qing Dynasty. The performers mainly sing and supplement this with narrations (J. Zhang 1984). Puyang Qinshu 濮陽琴書 is popular in the Puyang area that is on the border between Henan and Shandong provinces.

As for the Puyang Qinshu, in the Republican era, the studies and statistics are much simpler: Puyang Qinshu performers worshiped Quanzhen master QIU Changchun as the patriarch and belonged to the CHAI branch, one of the aforementioned four branches affiliated with the Longmen School; around the 1949 liberation, the relatively famous Qinshu performers were JIANG Hexiu 姜何修, ZHAO Jiaowen 趙教文, JING Yongfu 荆永福 (1922–1982), ZHU Yuanli 朱元立, MA Fengyun 馬鳳雲 and MA Shunqing 馬順卿 (*QYHB*, 2, p. 157; *QYHB*, 3, p. 93). They belonged to the He 何, Jiao 教, Yong 永 and Yuan 元 generations.

### 2.8. Yongcheng Da'nao

The name of this performing art comes from its instrument, *da'nao* 大鐃, which is usually used by a single person and is probably influenced by the Shandong Laozi. In the late-Qing and early Republic periods, the Da'nao genre was introduced to Yongcheng 永城

in Henan province by performer HAN Fengkui 韩鳳魁 (1862–1911), and then took root and flourished there. HAN adopted CHENG Xueshan 程學山 (1894–1960) as his disciple in Yongcheng and gave him the stage name Yuanfang 元方. Cheng had three disciples: MIAO Mingqing 苗明清, CHENG Mingyuan 程明元 and BIAN Mingkun 卞明坤 (*QYHB*, 3, pp. 32–34). In accordance with the Yuan 元 and Ming 明 generations in the lineage poem and the professional customs in local performing arts, we could presume that the successive order of Yongcheng Da'nao also belonged to that of the Quanzhen Longmen School.

We identified these eight local genres of the performing arts previously mentioned to have some kind of relationship with Quanzhen Taoism. Unlike the Taoist ballade and Shuoshu, other local performing arts emerged relatively late in time, mainly after the mid-Qing period. Because there is scarce historical documentation of these local arts, most of the performers we know, to date, were active in the first half of the 20th century, and most of their generation names were He 合, Jiao 教, Yong 永, Yuan 圓 and Ming 明. In regard to the characters that were used by their predecessors, we were unable to determined these, at present, due to the shortage of historical documents and oral accounts.

The reasons for the use of Quanzhen's institution by local performers are complex. The first reason to be taken into consideration is that the Quanzhen masters used the Taoist ballade to preach their doctrine during and after the foundation stage of Quanzhen Taoism. During the Jin 金 (1115–1234) and Yuan Dynasties, due to WANG Chongyang and his disciple, especially the Seven Veritables' painstaking effort, Quanzhen Taoism turned from an originally disclosed folk group into a national and institutional religious order that could compete with Zhengyi Taoism; at the same time, with regard to its method of propagation, Quanzhen Taoism paid special attention to local music and dance, which were popular during the Jin Dynasty, so as to be well-adapted to the local folk customs and attract public support (Z. Zhang 2011, pp. 113–15). Through this brief introduction, we can determined that these eight local performing arts have fused and influenced each other, for example, the Taoist ballade and the Zhuizi, the Shandong Laozi and the Yongcheng Danao. This influence may not only be in the form of performances, but also the institutions. The Taoist ballade clearly comes from Quanzhen, and, in turn, has influenced the other performing arts.

Secondly, some investigators have argued that the reason why local performers are affiliated with Taoism is closely connected to their social status in pre-modern society: "In the feudal society, to avoid being insulted or humiliated by the society, performers claim that they are descending from the lineage of immortals; born with natural air, walking in the three worlds (Heaven, Earth and Human World) and equipped with five elements (wood, fire, earth, metal and water), they claim Three Pures (Sanqing 三清) and Five Patriarchs (Wuzu 五祖) to be their ancestors" (*QYHB*, 2, p. 185). Even in the Republican era, the local artists' singing and narrating techniques were considered to be of a low standard and even difficult to be identified as one type of art: "a variety of people can be their audience, even women are regulars; most of the audience are the common people and the uncivilized soldiers who are trapped in though life and hard to hold their head high. So, a gentleman who is endowed with extraordinary talents and wealthy in knowledge or a westernized member of the Imperial Academy who can read English letters like 'A, B, C and D' would not go there" (L. Zhang 1936, p. 83).

During and prior to the Republican era, local performers indeed possessed an extremely low social status (L. Wang 1981; Me 2005). On the one hand, they were discriminated against as being uneducated; however, on the other hand, they mastered the performing arts that could entertain people. Spontaneously, they needed to justify the origin of their arts and Taoism became their first choice. Therefore, just as cited in the former paragraph, the performers claimed that they descended from the lineage of immortals and formed the Taoist orthodox teaching of "One Air (Yiqi 一氣) brings about Three Pures, Three Pures leads to Five Patriarchs, Five Patriarchs teaches Seven Veritables and Seven Veritables imparts Eight Lineages", in order to create their own sacred history. Most local

performers did not know more than their masters, or grand master in some cases; they quite clearly knew that they were the de facto successors of the Taoist immortals.

Lastly, the use of lineage poems is also a way for local performers to self-identify, and it is one of the most obvious signs of the close relationship between the performer's troupe and Quanzhen clergy. It not only expresses one's origin and transition, but also unites the master and disciple. Except for the storytellers who may have attended traditional private schools (*sishu* 私塾) for a few years, most performers were uneducated and had a low literacy level. The performers' education relied on their apprenticeships to their masters. Because the number of performers was small, each master took on fewer disciples, usually no more than ten, thus forming an extremely close relationship. This relationship was very similar to the master–disciple relationship of Quanzhen Taoists. Through the lineage poem, they could not establish only the generation between master and disciple, but could also identify with the genre of the performing arts to which they belonged.

To date, actors in Kunqu 昆曲 and other theatres also have their own lineage poems (Hu 2018); however, they are not those of the Quanzhen Longmen School. The phenomenon of the lineage poems of the Longmen School being used by the local performing troupes in Shandong and Henan provinces reflects the distinctive influence of Quanzhen Taoism, especially the Longmen School, in these two provinces. However, we should keep in mind that the Quanzhen influence is confined to the institutional organization; from the repertoire of Zhuizi (154 pieces in total), Shuoshu (73 pieces in total) and the Taoist ballade (65 pieces at least) preserved in the *Xiangguosi minzhong yule diaocha* (L. Zhang 1936, pp. 84–85, 104–105, 126–127), local actors preferred to perform and narrate popular legends and histories, especially the stories from the famous novel Sanguo yanyi 三國演義, which once mentioned a Taoist figure in the Taoist ballade *Hanxiangzi baishou* 韓湘子拜壽; however, it never dealt with Quanzhen teachings or practices. In other words, we may easily understand and categorize Confucian loyalty and filial piety, proselytized by the local performing arts; however, in the case of Quanzhen Taoism, in contrast with the so-called "Ch'üan-chen drama" of Yao Tao-chung (Yao 1980b) or the "Quanzhen plays" of David Hawkes (Hawkes 1981) in their respective studies on Quanzhen Taoism and the Yuan opera, we can easily find it institutionally replicated in the local performing arts.

### 3. Sacrificial Ceremony, the Acceptance of Disciples, and the Ever Spring Guild

Based on their strong social support for Quanzhen Taoism and emotional recognition of its identity, in the Republican period, various local performing arts recruited and taught their disciples in accordance with the *Hundred-Character Chart of Longmen School*, as previously shown, and, meanwhile, almost all genres performed the same ceremony for the sacrifice and acceptance of disciples, similar to those of Quanzhen Taoism. Moreover, for the purpose of survival in society, they set up the Ever Spring Guild to assist one another.

#### 3.1. Sacrificial Ceremony in Honor of the Patriarch QIU

Because of their affiliation with Quanzhen Taoists, these performers worshiped their own Taoist patriarchs, such as QIU Changchun and LÜ Dongbin 呂洞賓: "on the birth- or death-days of their patriarch, every performer would offer money to hold a memorial meeting to worship their own patriarch, and simultaneously suspend their business to spend time on holding a bustling festival" (L. Zhang 1936, p. 77). Indeed, in addition to their own patriarch, several performers also presented other special sacrificial rules. Taking the magicians in the Republican period as an example, in addition to worshiping their patriarch LÜ Dongbin, they would "burn incense, paper money and candles to pay respects to the Heaven on the 15th day of every lunar month" (L. Zhang 1936, p. 195).

Among the sacrificial ceremonies performed by different performing art troupes, the most influential is of course that of the Longmen School. Prior to the 1949 liberation, different kinds of performers, such as the Pingci performers who practiced martial arts and acrobatics, the Zhuizi and performers who wandered around without settling down anywhere, all assembled at Qiuzu temple 丘祖廟, located in South Mount Taishan temple street

in Kaifeng, every year on the birth- (the 19th day of the 7th lunar month) and death-days (the 19th day of the 12th lunar month) of the patriarch QIU (*QYHB*, 1, p. 60). Even the Patriarch Qiu temple fair attracted local artists from other provinces; according to the statements of XU Zhifa 徐治法, a Zhuizi performer from Fuyang county 阜陽縣 in Anhui 安徽 province: "In the past, people from Anhui province always came to Kaifeng to worship patriarch QIU on the 19th of the 7th lunar month" (*QYHB*, 1, p. 46). Indeed, in addition to the birth- and death-days, performers also visited the memorial tablet of the patriarch QIU during ordinary festivals to burn incense and prostrate themselves before the tablet (*QYHB*, 2, p. 253).

In the Republican era, according to the existing historical data, there were three leaders present during the sacrificial ceremony: the president (*huizhang* 會長), law-executor (*zhitangsi* 值堂司) and the manager (*zhuguan* 主管). The agendas of the sacrificial ceremony were: 1: worship the patriarch QIU Changchun; 2: preach ten precepts; 3: examine oneself before the patriarch and reflect on one's violation of precepts and the subsequent punished; 4: write a post (*xietie* 寫帖) for the new disciple; 5: "warm the birthday" (*nuanshou* 暖壽), that is, to express their sincerity, participants should burn incense and paper throughout the night and learn from each other by comparing their skills; and 6: prostrate oneself before the patriarch QIU Changchun, set off firecrackers and then end the sacrificial ceremony (Bianweihui 1995, pp. 509–11).

*3.2. Disciple Initiation Rites*

As previously mentioned, various local performing troupes associated themselves with the Longmen lineage, while during the actual transition, one performing troupe can be divided further into different branches, such as the ten branches of Taoist ballade performers, "ZENG 曾, CHAI 柴, YANG 楊, ZHANG 張, LIU 劉, GAO 高, QI 齊, LU 蘆, Sunzhao 孫趙 and Zhangzhao 張趙", and the eight branches of Henan Zhuizi: "Dazhang 大張, Xiaozhang 小張, Dahua 大花, Xiaohua 小花, SUN 孫, DONG 董, LIU 劉 and GAO 高." Out of the need for inheriting the traditions of performance and survival, local performers attached great importance to the master–disciple relationship and called the ceremony of initiating new disciples into the branch as "entry into the home" (*ru Jiamen* 入家門). The new performers could be formally identified only after participating in the disciple-initiation ceremony and, only after completing his apprenticeship could he then recruit and teach his own disciples. Otherwise, they would be labeled as an empty leg (*kongtui* 空腿) or wild breed (*yezhong* 野種) and were not allowed to make a living as a performing artist. In addition, other performers would not respect them and would even exclude them from their group by throwing away their performing instruments (*dian jiahuo* 掂傢伙) (*QYHB*, 3, pp. 205–7).

In fact, in order to be a Quanzhen cleric, formally accepted by both the Taoists and government, people had to undergo two separate rites of passage, ordination (*guanjin* 冠巾) and collective consecration (*shoujie* 受戒), of which the latter is more typical of Quanzhen Taoism (Goossaert 2007, pp. 140–52; Goossaert 2013, p. 36). During the ordination rite, one has to write a post, clearly showing his firm desire to leave the family (*chujia* 出家); during the collective consecration ceremony, at least eight famous Taoists serve as the guarantors and monitors witnessing the entire ceremony (L. Gao 2018, pp. 110–13, 149–53; Peng 2021, pp. 103–10). Similarly, when a local actor accepts a disciple, many of these factors appear in the rite and, more importantly, the frequent reference to and presence of Wang Chongyang and his famous disciples, the Seven Veritables, made the rite more similar to a Quanzhen initiation event. For example, when the Shandong Kuaishu performer GAO Yuanjun formally acknowledge QI Yongli as his master, he burned incense in front of the memorial tablet, "Heaven, earth, emperor, parents and master (Tian di jun qin shi 天地君親師)", located in the middle of the hall; he then prostrated himself before his master, listening to his master announcing: "I, a little Taoist priest in blue clothes, go down the hill under the immortal master's order; today is no other than an auspicious day, and I will transmit the Tao for the immortal master."[7] Some other local performing arts more obviously

evoke Quanzhen lore. For instance, when a Guangzhou Dagu troupe holds its ceremony for accepting new disciples, a new performer presents a special post of initiation (*baishi tie* 拜師貼) to his future master. Such a post of initiation often traces its sacred history during the introductory stage (*tietou* 貼頭):

> Since Pangu created the world from the chaos of the universe, heaven, earth, emperor, parents and master were the superiors;
>
> 混沌初開盤古分, 天地君親師為尊；
>
> Primordial Patriarch Hongjun laozu set up the great Longmen and dispatched WANG Chongyang to come down to Earth to save and enlighten the ordinary people;
>
> 鴻鈞老祖立下大龍門, 差派王重陽下凡來度人；
>
> WANG Chongyang adopted one hundred disciples in total, and seven of them attained Tao: QIU, LIU, TAN, MA, HAO, WANG and SUN.
>
> 徒弟一百整, 得道有七人：邱、劉、譚、馬、郝、王、孫.
>
> Having arrived at the Red Cloud Temple, they finally settled down there;
>
> 到了紅雲寺, 師徒才安身；
>
> Later, WANG Chongyang died and patriarch QIU kneeled down in front of the gate (of Red Cloud Temple);
>
> 貼生喪了命, 邱祖去跪門；
>
> The (seven) disciples searched all over for the Zhouzu ground, and then safely intered their master;
>
> 尋找周祖地, 葬安師父身；
>
> Having observed a one-hundred-day mourning period for their master, they then set up their schools separately;
>
> 守孝百天整, 各立各的門；
>
> Patriarch QIU created the Longmen School, and was therefore called the Old Superior Worthy.
>
> 邱祖龍門派, 長稱老先尊. (QYHB, 3, p. 38)

In this brief introduction, Honjun Laozu is not a Quanzhen immortal, but the supreme deity in the *Investiture of the Gods* (Fengshen yanyi 封神演義). Local artists often sang and narrated this famous divine novel (Meulenbeld 2015). In the Republican era, local art performers' initiation post for acknowledging someone as a master also included the main text, regulations, performers' one-hundred-generation characters, the signature of the guarantor and recommender masters and the date. We should note that the one-hundred-generation poem is the lineage poem of the Longmen School in Quanzhen Taoism (*QYHB*, 3, p. 228).

However, the acknowledgment of someone as the master is not performed at random; it also involves many ritualistic and economic requirements. For example, the master-acknowledgment ceremony requirements for a Henan Zhuizi performer are as follows:

1. Parents, brothers and sisters cannot be acknowledged as the master, only another non-kin person can.
2. When acknowledging the master, it is necessary to have recommender (*yinjianshi* 引薦師) and Taoist (*daoshi* 道師) masters.
3. On that day, one should invite all the participants to a feast to let them know that you have acknowledged someone as your master.
4. It is necessary to write a master-acknowledgment post and invite someone to serve as the guarantor master (zuobaoshi 作保師).
5. Having finished the three-year apprenticeship, one should perform the Zhuizi for one year for the master and donate all the earnings to the master as a repayment (L. Zhang 1936, p. 74).

Out of these five requirements, the 3rd and 5th items are derived from performing troupes' common habitus; the 2nd and 4th items have a distinct ritual significance: the presence of the recommender, Taoist and guarantor masters shows that the master-acknowledging ceremony of Quanzhen Taoism has an obvious effect on the local performing arts.

More typical is the master-acknowledging ceremony of Shandong Laozi. According to its customs, in addition to the three masters previously mentioned and his fellow apprentices, during the ceremony, one should also submit to the disciple post (*mensheng tie* 門生貼) and perform a salutation to the master; later, the master would give him a stage name, according to the Longmen lineage poem, add his name to the disciple register, teach him the rules and preach the moral rules concerning acting.[8] Based on these main factors and procedures, we can easily observe that the performer's ceremony for acknowledging someone as the master is, to a certain degree, similar to the ordination and collective consecration rites of the Quanzhen Longmen School.[9]

### 3.3. The Ever Spring Guild

In the former section, we mentioned that local performing actors hold a sacrificial ceremony for the patriarchs QIU Changchun and LÜ Dongbin on a special date. In reality, these Quanzhen celebrities played a role in the local actors' daily lives. For example, LÜ Dongbin was viewed as the patron god of prostitutes for his legendary flirting with a white peony (Katz 1996; P. Liu 1996), and another Quanzhen patriarch, LIU Chuxuan, was treated similarly because he was said to cultivate himself in a brothel (Qin 2023). In regard to QIU Changchun, he was perceived as the founder and patron god of jade craftsmen and their guild, which we refer to again later in the study, because people believed that QIU Changchun taught the poor to transform stone into jade (Y. Li 2002, pp. 732–733). Whether prostitutes or jade craftsmen, they were considered low social classes, and by founding an association or a guild, they could unite and support each other in the face of economic depression or social changes. Likewise, this also occurred in the local performing arts during the Republican era.

During the Republican period, two main types of performers' guilds existed in Henan province: one was the Three Sovereigns Guild (Sanhuang hui 三皇會), where blind and Sanxianshu 三弦書 (narrations and singing with *sanxian* as an accompaniment) performers dominated, and the other was the Ever Spring Guild, which was dominated by Taoist ballades and story-telling performers. In his book, the *Investigation of Popular Entertainment in the Xiangguo Temple*, ZHANG Lüqian 張履謙, on the basis of the sociological survey data from Republican Kaifeng, stated that the Ever Spring Guild was a popular organization consisting of people who told stories, sang Taoist ballades and Zhuizi, performed conjuring tricks and performed acrobatics. In 1936, the president of the Ever Spring Guild was the Taoist ballade performer MA Lichun and the vice president was Zhuizi performer ZHANG Mingliang (L. Zhang 1936, p. 73).

In fact, the Ever Spring Guild, during the Republican era, was not established only in Kaifeng, but also in Zhengzhou 鄭州 (Henan province), Ji'nan and Peking. Moreover, participants were not limited to the local performing arts discussed in this paper, but also assumed a wide variety of popular occupations; in his *Stories of rivers and lakes* (Jianghu congtan 江湖叢談), LIAN Kuoru 連闊如 stated that all performers and retailers could join in the Ever Spring Guild, for example, people who told fortunes by reading faces; performed Chinese martial arts and acrobatics; sold liniments, eye ointments, plasters, medicine for toothaches, afrodyn, knives and scissors, needles and combs; performed and sold conjuring tricks; sang Dagu Shu; told story recited to the rhythm of bamboo clappers; commented on the histories; performed a comic dialog and pedicures; sold medicine for simian sarcoma, other special medicines and folk prescriptions; healed venereal diseases; performed with monkeys and other animals; performed raree shows; sold medical sugars and ratsbane; and performed the circus.[10]

By listing the popular occupations, we could observe without any difficulties that these people lived at the grassroot level of society. As they faced daily pressures from their



difficult lives, they needed to set up or join in the Ever Spring Guild to assist each other. Similar to the Ever Spring Guild, they also united and collectively created a legendary tradition about the "thirteen boards" (*shisankuai ban* 十三块板), which assembled all the disadvantaged and local artists to form a family, "whether knowing each other or not, once heard someone saying 'let us see on the thirteen boards', they will treat each other as a family" (*QYHB*, 2, p. 251). To put it another way, to "assist each other" or "treat each other as a family" is the practical reason for founding or entering the Ever Spring Guild.

The Ever Spring Guild had a president and a vice-president. Both of them were elected by the performers. "The president should have rich experiences, earn more money than the others, behave honestly, have the courage to solve tough problems, work hard, be willing to sacrifice himself and be able to mediate disturbance and dispute. Only in this way, people would respect him, and move under his orders when thing happens." Moreover, the presidency had no limitations, "only under the condition of having big faults, incurring the wrath of the public or resigning voluntarily, the president could be changed." The main responsibilities of the president were divided into two categories: internal and external. Internal responsibilities included accommodating performers when hosting temple fairs, establishing performance areas and punishing performers who violated guild regulations, etc.; the external responsibilities included assisting the local gentry to host temple fairs and markets, taking care of routine temple-fair matters and maintaining performers' economic interests (Lian, pp. 4–7).

It should be noted that the Ever Spring Guild was an unofficial, fluid organization. As it was freely composed of local performers, it was quite weak at practically disciplining local artists. Taking Xiangguo temple, for example, in the Republican era, "the Ever Spring Guild just drew up the guild regulations and conventions and took care of routine recruiting matters. As for the problems of choosing performing area or performing what kind of genre, the Guild is totally laisser-faire: performers can randomly perform or sing what they want." However, when addressing the collective crisis, for example, in 1927, when General FENG Yuxiang 馮玉祥, in the name of rectifying social order, planned to prohibit the popular arts, such as telling stories and singing, performers of various local arts could voluntarily assemble and help each other (*QYHB*, 1, p. 229). In other words, the Ever Spring Guild, founded in the name of patriarch QIU Changchun, formed a common identity and served as a clarion call for them to unite and organize collective activities.

Whether from the sacrificial ceremony, master-acknowledgment ritual or Ever Spring Guild in the name of QIU Changchun, we can easily identify Quanzhen lore in the local performing arts. Moreover, we can alternatively claim that Quanzhen Taoism, as a completely institutionalized religious order, is the model to imitate for local performing artists. However, we should note that, at the same time, imitation is never duplication; when adopting the Quanzhen institution, local artists did make some deviations and variations for their individual needs.

## 4. Deviation and Variation

The transmission of the Longmen lineage poem into the local performing arts is closely related to the preaching methods of Quanzhen Taoism. At the beginning of Quanzhen Taoism, Taoists often preached by singing Taoist ballades (Z. Zhang 2008; Liao 2022). It was during this preaching process that Quanzhen Taoists may have brought their way of teaching into the local performing arts troupe. As a result, the performers of Taoist ballades also used the Longmen lineage poem and, eventually, they expanded from Taoist ballades to other performing arts (*QYHB*, 3, p. 13).

However, while they adopted the Quanzhen Taoist lineage poem as an institutional form, the local artists considerably changed Quanzhen Taoist history and rewrote the Longmen lineage poem with different characters. This may have been due to the poor education of the performers. Although they might have known or heard about the immortals and legends of some Quanzhen patriarchs, they did not precisely understand Quanzhen doctrines, ideals or its history of transmission, which was only accessible to Quanzhen clerics.

Additionally, living a difficult lives, local performers of different generations might become sworn brothers or get married, which also disrupts the regular successive sequence of the original Longmen lineage poem. Therefore, compared to the Quanzhen institution, and if we take that as a standard, the organizational structure of the local performing arts presents many deviances and variances; thus, they seem to be similar, but are actually quite different.

### 4.1. Quanzhen History

We already stated that the Quanzhen history we know, to date, is constructed by Quanzhen adherents, especially Taoists of the third generation, to identify the patriarchs existing before WANG Chongyang and classifying the Seven Veritables as a group (Marsone 2001; G. Zhang 2008; W. Zhao 2010). Whether the history is true or not, we must perceive it as a fact that Quanzhen Taoism indeed created a sacred history, namely, the Five Patriarchs and the Seven Veritables, and this was widely accepted as the standardized formula by Quanzhen Taoists from the mid-Yuan dynasty onwards (Jing 2012). Since all the local performing arts we discussed here affiliate themselves with the Longmen School, when tracing their own history, they also needed to deal with the origin and development of Quanzhen Taoism. However, just as we showed in other sections of this paper, most of their narratives are quite different from the standard Quanzhen history during the Yuan dynasty, where many Quanzhen immortals and masters are replaced by mythical and imaginary Taoist figures, or many non-Quanzhen immortals and masters only appear in Quanzhen history without providing any explanations.

Taking the origin of Guangzhou Dagu as an example, according to local performers' oral accounts, "(Guangzhou Dagu) was first performed by the earliest Taoist ancestor Hongjun laozu, then he hands down his knowledge and skill to ZHANG Guolao 張果老, one of the Eight Immortals (Baxian 八仙), XU Maogong 徐茂公 of the Tang Dynasty and WANG Chongyang of later dynasty, etc." (*QYHB*, 3, pp. 37–38). WANG Chongyang had one hundred disciples and seven of them attained the Tao, namely, the "Seven Veritables": QIU, LIU, TAN, MA, etc. Then, QIU Changchun became the successor and continued to recruit and teach eight people, GAO, GUI, CHAI and ZHANG, etc., namely, the "Eight Lineages". This was the outcome of the saying "Seven Veritables, Eight Lineages and One Hundred Branches", and from where the introduction of the post for acknowledging someone as the master originated. Hongjun laozu, ZHANG Guolao, XU Maogong and WANG Chongyang are either Taoist immortals or real Taoist figures; however, their successive transitions from master to disciple in Guangzhou Dagu were very different from the various hagiographical accounts presented in Quanzhen literature.

We can more clearly observe this fabrication from the aforementioned communal pantheon that recounts their origins and is prevalent in various local performing arts: they are all affiliated with the lineage of immortals, descending from the natural air, and taking Three Pures and Five Patriarchs as their ancestral root; Three Pures refers to Yuanshi Tianzun 元始天尊, Lingbao Tianzun 靈寶天尊 and Taishang Laojun 太上老君; Five Patriarchs refers to Donghua dijun LI Tieguai 東華帝君李鐵拐, Jinghua dijun LÜ Chunyang 警化帝君呂純陽, Shunhua dijun CAO Guojiu 順化帝君曹國舅, Chuandao dijun ZHONGLI Quan 傳道帝君鐘離權 and Fuzuo dijun LIU Haichan 浮佐帝君劉海蟾, all belonging to the Eight Immortals; In the Song Dynasty, CAO Guojiu was imprisoned as he illegally favored his guilty brother. After his release, CAO traveled to a mountain to cultivate Tao, named his branch the "Chongyang pai" 重陽牌, of which Chongwang means a return to life, and transmitted "Chongyang pai" to seven disciples, namely, QIU, LIU, TAN and MA, etc.; then, the seven disciples set up eight branches in the local performing arts, such as Longmen and Huashan. This was their sacred tradition: "One Air brings about Three Pures, Three Pures leads to Five Patriarchs, Five Patriarchs teaches Seven Veritables and Seven Veritables imparts Eight Branches" (*QYHB*, 2, p. 185). However, this complete and standard history, re-written by local performers, is a hybrid; WANG Chongyang, the

founder of Quanzhen Taoism and a real figure in history, is excluded for no reason and exists as a branch name.

Compared to classical Quanzhen history, such as the *Record of the True Line of Transmission of the Golden Lotus School* (Jinlian zhengzong ji 金蓮正宗記, DZ173) and *Portraits and Biographies Concerning the Origin of the Master of the True Line of Transmission of the Golden Lotus School* (Jinlian zhengzong xianyuan xiangzhuan 金蓮正宗仙源象傳, DZ174), we can easily observe that there is a great difference between Quanzhen history compiled by Quanzhen Taoists themselves and the sacred history recounted by local performers. As previously mentioned, the reason why these popular performers refer to Quanzhen history is because they want to borrow and make use of the reputation of Quanzhen Taoism in the Jin and Yuan dynasties, and then construct their own truth and orthodoxy. From of this consideration, they inherited the Taoist pantheon, such as Three Pures and Five Patriarchs, already existing in Taoist tradition, and, alternatively, transplanted and grafted this pantheon into their own traditions and branches, such as the ten branches of the Taoist ballades and the eight branches of Henan Zhuizi. In short, they changed the history unanimously accepted by Quanzhen Taoists and constructed their own sacred history by using Quanzhen masters and legends to describe their origins according to their own circumstances and needs.

On first sight, these variations in or deviations from the standard history of Quanzhen patriarchal succession may seem random or capricious, lacking the necessary interpretations of these changes from the performer's part. However, these deviant histories of the local performers may stem from their local circumstances and needs. For instance, Hongju laozu might be more familiar to the locals than QIU, LIU, TAN and MA. In other words, these local ballad singers and storytellers were trying to adapt to the tastes and preferences of their audiences. Their reconstructed histories, though deviating from standard Quanzhen history, were not random at all but, rather, they were reasonable and calculated adaptations.

We also need to consider another possibility, as previously stated: the identity exchange between Quanzhen Taoists and local performers in late-Imperial China. Many Quanzhen Taoists eventually became professional performers and, vice versa, many performers were household Quanzhen Taoists. According to the fieldwork of some contemporary scholars, this was indeed the case. Stephen Jones' research, for example, determined that, when clerics were expelled from the temples or returned to lay life, they took on the professions of performers or trained lay people in the arts ([Jones 2011](#)). This was not deception; it was a real lineage succession within which lifestyle and profession evolved. From this, we can speculate that a similar transmission may have existed in the northern society in the Republican period. However, in reviewing the local sources of Shandong and Henan that we have observed, to date, concerning the Republican period, we did not obtain any obvious evidence.

*4.2. Birth-/Death-Days of Patriarch QIU Changchun and His Disciples*

During the Republican era, the Ever Spring Guild, voluntarily organized by all kinds of local performers, would regularly hold gatherings to commemorate their common patriarch, QIU Changchun. Undoubtedly, patriarch QIU's birth- and death-days became the most important dates for gatherings. However, the dominant saying in Republican performing arts genres is that the birthday of patriarch QIU is on the 19th day of the 7th lunar month and the death date is the 19th day of the 12th lunar month; according to Quanzhen history, neither of these two dates conforms to historical fact. In fact, QIU's birthday is on the 19th day of the 1st lunar month and the 9th day of the 7th lunar month. The underlying cause for the change in the sacrifice date is undecipherable, at present. This may be because the two dates were well-established by local market fairs or temple festivals. Moreover, the two dates were the birth or death dates of an important performer in history and were then used to mark the sacrificial rituals of the ancestors of the genre.

At the same time, for their own needs, local performers made some changes to patriarch QIU's disciples. QIU Changchun adopted five disciples in total: the first sang opera,

the second told stories, the third wove bamboo baskets (*luo* 籮), the fourth repaired win-
nowing pans (*boji* 簸箕) and the fifth repaired bamboo baskets (*QYHB*, 2, p. 251). However,
according to *The History of the Taoist School founded by (Qiu) Changchun* (Changchun daojiao
yuanliu 長春道教源流), none of QIU's 48 first-generation disciples engaged in any of these
five kinds of occupations (M. Chen 1975). The most reasonable explanation for this alter-
ation is that these five, disciple craftsmen attributed to patriarch QIU were representatives
of the people who actually lived in grassroot society and joined the Ever Spring Guild. Sim-
ilar patterns can also be observed in jade craft guilds in Peking and in other North China
cities and townships where local jade craftsmen and merchants adopted QIU Changchun
as their patron god; however, traditional Quanzhen history never mentions QIU as having
anything to do with jade craft or business. Local circumstances and needs are crucial to
these apparent or random choices in reshaping history.

*4.3. Lineage Poem and the Successive Relationship of Master and Disciple*

In the Republican era, numerous local artists named their disciples in accordance with
the lineage poem of the Longmen School of Quanzhen Taoism; however, with regards to
each character, numerous differences between the lineage poems known to performers of
different genres and that of the Quanzhen Longmen School recorded in the *General Register
of Lineages Revealed by Various Veritables*, which are presented in the following chart one by
one, are evident (Table 3):

**Table 3.** Lineage poems of different genres.

| School/Branch | 派诗 Lineage Poem | |
|---|---|---|
| 白雲觀<br>White Cloud<br>Monastery | 道德通玄靜 真常守太清 一陽來複本 合教永圓明<br>至理宗誠信 崇高嗣法興 世景榮惟懋 希微衍自寧<br>(Oyanagi 1934, p. 97; K. Wang 2009, p. 65) | |
| | DAO DE TONG XUAN JING<br>YI YANG LAI FU BEN<br>ZHI LI ZONG CHENG XIN<br>SHI JING RONG WEI MAO | ZHEN CHANG SHOU TAI<br>QING<br>HE JIAO YONG YUAN MING<br>CHONG GAO SI FA XING<br>XI WEI YAN ZI NING |
| 道情<br>Taoist ballade | 道德通玄靜 真常守太清 一陽來複本 合教永圓明<br>至理宗誠信 崇高嗣法興 世景榮惟懋 希微衍自寧 (*QYHB*, 3, p. 226) | |
| | DAO DE TONG XUAN JING<br>YI YANG LAI FU BEN<br>ZHI LI ZONG CHENG XIN<br>SHI JING RONG WEI MAO | ZHEN CHANG SHOU TAI<br>QING<br>HE JIAO YONG YUAN MING<br>CHONG GAO SI FA XING<br>XI WEI YAN ZI NING |
| 河南墜子<br>(張明亮)<br>Henan Zhuizi<br>(Zhang<br>Mingliang) | 道德通玄靜 真常守 泰 清 一陽來複本 合教永 元 明<br>至理宗誠信 崇高嗣法興 世景榮 為 懋 希 征 衍自寧 (L. Zhang 1936,<br>reprint in 1989, p. 75) | |
| | DAO DE TONG XUAN JING<br>YI YANG LAI FU BEN<br>ZHI LI ZONG CHENG XIN<br>SHI JING RONG WEI MAO | ZHEN CHANG SHOU TAI<br>QING<br>HE JIAO YONG YUAN MING<br>CHONG GAO SI FA XING<br>XI WEI YAN ZI NING |

**Table 3.** *Cont.*

| School/Branch | 派诗 Lineage Poem | |
|---|---|---|
| 河南墜子 (柴門)<br>Henan Zhuizi<br>(Chai branch) | 道德通玄靜 正乾 守太清 義 陽來 富 本 合教永 元 明<br>志利忠 誠信 慶 高 四 法興 習景雍偉茂 勝玉衍子平 (*QYHB*, 3, p. 206) | |
| | **D**AO DE TONG XUAN JING<br>**YI** YANG LAI **FU** BEN<br>**ZHI LI ZHONG** CHENG XIN<br>**XI JING YONG WEI MAO** | **ZHENG QIAN** SHOU TAI QING<br>HE JIAO YONG **YUAN** MING<br>**QING** GAO **SI** FA XING<br>**SHENG YU YAN ZI PING** |
| 評詞說書<br>Pingci Shuoshu | 道德通玄靜 南 常守太清 一陽來 富 本 何 教永 遠 明<br>至 禮忠 誠信 崇高嗣 發 興 世景榮惟懋 希微衍自寧 (*QYHB*, 1, pp. 60–61) | |
| | DAO DE TONG XUAN JING<br>YI YANG LAI **FU** BEN<br>ZHI **LI ZHONG** CHENG XIN<br>SHI JING RONG WEI MAO | **NAN** CHANG SHOU TAI QING<br>**HE** JIAO YONG **YUAN** MING<br>CHONG GAO SI **FA** XING<br>XI WEI YAN ZI NING |
| 山東落子<br>Shandong Laozi | 道德 同先慶 鎮鍵 守太清 意言 來 富 本 和 教永 元 明<br>志禮忠誠信 從 高 士發 興 始宗龍為廟 西湖岩子寧 (Li Li 2016)[11] | |
| | DAO DE **TONG XIAN QING**<br>**YI YAN** LAI **FU** BEN<br>**ZHI LI ZHONG** CHENG XIN<br>**SHI ZONG LONG WEI MIAO** | **ZHEN JIAN** SHOU TAI QING<br>HE JIAO YONG **YUAN** MING<br>**CONG** GAO **SHI FA** XING<br>**XI HU YAN ZI NING** |
| 光州大鼓<br>Guangzhou Dagu | 道德通玄靖 遵 常守太清 陰 陽來 複本 和 教永 元 明<br>智禮忠 誠信 崇高 賽發 興 詩經 榮 易茂 喜為宴子林 (*QYHB*, 3, p. 40) | |
| | DAO DE TONG XUAN JING<br>**YIN** YANG LAI FU BEN<br>**ZHI LI ZHONG** CHENG XIN<br>**SHI JING** RONG **YI MAO** | **ZUN** CHANG SHOU TAI QING<br>**HE** JIAO YONG YUAN MING<br>CHONG GAO **SAI FA** XING<br>**XI WEI YAN ZI LIN** |

From this chart, we can observe that there are numerous differences between the characters; however, their pronunciations are almost the same. There are three possible reasons for the differences in the lineage poems. The first is that different versions of the Longmen lineage poems are created during the process of transmission. Even within the Longmen lineage, varying versions of the lineage poems have existed since the mid-to-late Ming dynasty. Various Quanzhen Longmen histories also recorded various genealogical characters among the local Longmen lineage and sub-lineages in different regions in north and central China (Esposito 2004; G. Zhang 2011; X. Zhang 2013; F. Zhang 2018). The second is that, during the process of transmission, due to the accents in different regions and the lack of written records, lineage poems can easily change during their dissemination, with similar pronunciations, but different characters, for example, the characters Zhi 芝/ 治 and Li 禮/ 理 in the names of Zhuizi performers in Xiangguo temple, previously mentioned (W. Guo 2017; Liu and Gao 2020). The third is the inadequate understanding of the Taoist doctrine and philosophical meanings by the performers, for example, the phrase " 希微衍自寧", which means "it is in what is held and subtle that one finds peace within oneself" (Herrou 2005, p. 316), in the poem of the White Cloud Monastery's character is expressed as " 西湖岩子寧" in Shandong Laozi and " 喜為宴子林" in Guangzhou Dagu, none of which can be understood or explained by local performers of these two genres. The words XI 希, WEI 微 and NING 寧 all frequently appear in the *Daodejing* 道德經, the oldest and most famous Taoist text. According to Yoshioka Yoshihiro, lineage poems can disclose the lineage's independence and doctrinal differences and express the lineage founder's understanding of his own enlightenment (Yoshioka 1979, p. 231). Facing the status quo, despite the similar pronunciations, the doctrinal differences and philosophical meanings are getting lost in translation.

However, when it comes to different branches using the same lineage poem, things becomes much more complicated: performers in the same generation can belong to different branches; performers in different generations can become brothers or can even marry each other. In other words, although local performers have their own stage name, in strict accordance with the lineage poem, they can also easily break this order of succession. Thus, "when trying to find out a performer's successive relation, we should not only know his stage name, but also ask his background; only in this way, could we then determine to which generation, which branch and which lineage he belongs" (Ma 1989, p. 131).

Local artists constructed their own sacred history by virtue of Quanzhen Taoism and, at the same time, changed parts of it according to their practical needs and personal understanding of it. Based on these statements, we can argue that, in the process of imitating the Quanzhen Taoist institution, the local performing arts, more or less, deviated and varied from tradition. From this arbitrariness, we can sense the local artists' desire to improve their social status and efforts to institutionalize their organization. In other words, this "seemingly alike but actually different" concept is exactly what popular performers present in their pursuit of institutional charisma issuing forth from Quanzhen Taoism.

## 5. Conclusions

In the Republican era, and possibly as early as the mid-Qing dynasty, when the majority of local performing arts emerged in China, out of the need to improve and upgrade one's social status, various local performing arts extensively borrowed and imitated the Quanzhen Taoist institution, including the theatrical branches, relationship of master and disciple and sacrificial and master-acknowledgment ceremonies, from which we can easily discover and identify the influence of Quanzhen Taoism. However, this imitation or borrowing is not achieved in a precise fashion, but with deviations and variations to different degrees, with some aspects becoming extremely different from the original concept: Quanzhen Taoism is more an ideal than a reality. In other words, Quanzhen Taoism greatly influences the local performing arts because of its the institutional charisma, while, during its flow into the other popular culture traditions, Quanzhen history was also considerably changed or reconstructed by local performers.

Indeed, taking the local performing arts as an example to describe the flow of institutional charisma is an interesting and important subject in Quanzhen studies[12]. Just as we showed in the case studies conducted by LAI Chi-tim 黎志添 on Guangdong Taoism and LI Dahua 李大華 on Hong Kong Taoism, Quanzhen is a tradition with a long history and high reputation; therefore, local Taoist sects were eager to assimilate and incorporate certain aspects (Lai 2007; D. Li 2018). At present, the Quanzhen Taoist tradition is still alive as the field research conducted in Hunan province indicates that a local Taoist sect in the Yuxu Gong 玉虚宫 in Xinhua 新化 county utilizes religious names while performing the liturgy (Tian 2023). However, on the basis of all these regional and case studies and by describing the transformations of the Quanzhen institution, we can reconsider the following two questions.

First, the place of Quanzhen Taoism in popular traditions. Rolf Stein argued that there is a "ceaseless dialectical movement of coming and going" between Taoism and other popular traditions:

> The priests of the great religions willingly adopt popular elements and adapt them to their system and nomenclature. Inversely, when popular milieux are confronted with one or more of the great religions, they easily fall subject to the prestige of the latter, and they too proceed to assimilate, make identifications, and become syncretic; or they may even replace old traditional forms with new ones borrowed from a great religion because these have more prestige. (Stein 1979, pp. 53–54)

Undoubtedly, from the description and analysis presented in the above-three sections of this paper, we can observe that this kind of dialectical movement is precisely what occurred between Quanzhen Taoism and the local performing arts in the Republican era.

However, it is worth noting that, until the Republican era, Quanzhen Taoism was established as a legal and complete institution for over seven hundred years. Therefore, in its interaction with the local performing arts, Quanzhen Taoism mainly focused on its institutional output; on the contrary, the local performing arts prioritized the imitation and absorption of the Quanzhen institution. In other words, by virtue of its unmatchable institutional charisma, Quanzhen Taoism dominated this interaction and considerably changed the popular culture diffused into quasi-institutional traditions. Therefore, compared to other types of popular cultural traditions, we can argue that Quanzhen Taoism plays a dominant role in society and exists as a powerful tradition.

Second, the transmission of Quanzhen Taoism in local society.

From extant Quanzhen documents, the main way to transmit Tao is the secret formulae (*mijue* 密訣) passed from master to disciple, and the main locale for transmitting Tao is monasteries and temples. Additionally, around these monasteries and temples, Quanzhen Taoism can organize a series of lay groups and guilds cultivating Tao, thus spreading Quanzhen teachings to local societies. However, in their pursuit of institutional charisma, lay groups and guilds can, on the one hand, imitate the Quanzhen institution and, on the other hand, make some appropriate adjustments and alterations, so as to survive and spread themselves in society.

Judging from the local performing arts during the Republican era, we understand that this newly adapted path for transmission, even though the succession from master to disciple remains fundamental and necessary, is not related to Quanzhen monasteries and temples: it has deviated from Quanzhen Taoism and even lost its original religious meaning. Local performers can still use the lineage poem to name their disciples; however, as for the religious or philosophical meanings of the poem itself, none of the performers we encountered in the documents or investigated in reality could explain, or even be told by their masters how to explain, them. These facts, possibly slightly frustrating for a Quanzhen scholar at first sight, still provide solid and concrete evidence of Quanzhen Taoism's enduring legacy in society and culture in late-Imperial and early Republican China.

**Author Contributions:** Writing—original draft preparation, G.Q.; writing—review and editing, W.Z. All authors have read and agreed to the published version of the manuscript.

**Funding:** This research was funded by The National Social Science Fund of China 國家社會科學基金 (Mingqing quanzhenjao de lishi chongshu jiyu minjian zongjiao guanxi yanjiu 明清全真教的歷史重述 及與民間宗教關系研究), grant number: 20BZJ042.

**Informed Consent Statement:** Informed consent was obtained from all subjects involved in the study.

**Acknowledgments:** Vincent Goossaert and LIU Xun have read the awkward manuscript, and give many quite important suggestions and comments, for which we are extremely grateful.

**Conflicts of Interest:** The authors declare no conflict of interest.

## Notes

[1] In 1927, Feng Yuxiang 馮玉祥 abolished temples in Henan, expelled monks and turned Xiangguo Temple into a market, where lots of performers came to live and perform. ([Zhong 1995](#)).

[2] For a description of oral accounts and its importance in religious studies, see ([Welch 1967](#), chap. IX, X, XI).

[3] See the Shandong Laozi.

[4] A single-headed hand drum with body made from a bamboo tube, used as an accompanying instrument in Daoqing narrative singing.

[5] There are much research on Taoist ballade, see ([Wu 1997](#); [Ye 1975](#), pp. 625–89; [Idema 2016](#)). On the definition of Taoist ballade and its connotation, see also Zhu Quan 朱權, *Taihe zhengyin pu* 太和正音譜. 1:42a

[6] About Xu Liping's performance and transmission, also see ([Li 2016](#)).

[7] 小小道童身穿藍, 我奉仙師下高山. 今逢黃道是吉日, 我替仙師把道傳. ([Jiang 2009](#), p. 126)

[8] *QYHB*, 2, p. 156. About the performing rules and acting morality, see *Zhongguo quyizhi henan juan*, pp. 506–507.

[9] About the Quanzhen rites of ordination and collective consecration, see (Y. [Chen 2003](#), pp. 21–27; L. [Gao 2018](#), pp. 105–10).

10 　算卦相面的, 打把式卖艺的, 卖刀疮药的, 卖眼药的, 卖膏药的, 卖牙疼药的, 卖壮药的, 卖刀剪的, 卖针的, 卖梳篦的, 变戏法的, 卖戏法的, 唱大鼓书的, 唱竹板书的, 说评书的, 说相声的, 修脚的, 卖猴子药的, 卖药子的, 卖偏方的, 治花柳的, 耍猴儿的, 玩动物的, 拉洋片的, 卖药糖的, 卖耗子药的, 跑马戏的等等. (Lian 2010, p. 4).

11 　In our fieldwork, we find that the lineage poem recorded by Xu Liping, under the name of xueyi menhu 學藝門戶, is another version, which is as follows: 道德同先庆 镇健守太清意彦来副本 和教永元明 知礼忠成信 从高士法兴始总龙委庙 西湖岩字宁 DAO DE TONG XIAN QING ZHEN JIAN SHOU TAI QING YI YAN LAI FU BEN HE JIAO YONG YUAN MING ZHI LI ZHONG CHENG XIN CONG GAO SHI FA XING SHI ZONG LONG WEI MIAO XI HU YAN ZI NING

12 　About other similar research subjects in Quanzhen Taoism, see (Dong 2003, pp. 578–611; Gao 1997, p. 128; Jones 2010, pp. 85, 88–89, 131–32, 147, 168–69).

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
