# Peer review of "The Flow of Institutional Charisma: Quanzhen Taoism and Local Performing Arts in Republic Shandong and Henan"

_religions, doi:10.3390/rel14050560_

Round 1
Reviewer 1 Report
See attached Word file report.

Reviewer 2 Report
In the abstract: "Quanzhen Taoism and its relationship with local performing arts is a very interesting subject... (Please avoid the word "interesting" because it does not tell us what this article offers.
The history of Quanzhen Taoism's roots in Republican Shandong and He'nan with its eight genres of local performances arts contributes vital information about Quanzhen Taoism.
The work is often missing the article "the."
Please reword the section on page 7, the last paragraph, which describes the performers singing and narrating techniques which are considered to be "low tastes" because it sounds extremely important but yet unclear. I think it will be very significant to readers. How was the aim to upgrade social status?
Would it be possible for you to add a section on which Taoist beliefs were conveyed in the ballads for example?
Reviewer 3 Report
The manuscript describes the relationship between Quanzhen Daoism and local art performers in Shandong and Henan during Republican China (1912-1949). It argues that local performers borrowed Quanzhen elements to raise their own social status, but that within this process they also adapted some Daoist elements according to their own needs. The manuscript is an important case study of the dynamic interaction between Daoism and local societies in Republican (1930s) and Communist China (1950s and 1980s).
General remarks:
1) The manuscript is predominantly focused on the Republican period, yet it is entirely written in the present tense. If there is a specific reason to utilise the present tense, it should be explained at the beginning of the text or in a footnote. However, since the manuscript is based heavily on written sources from the 1930s, 1950s, and 1980s – and only mentions interviews with present-day performers once or twice – it would be more comprehensible for readers if the past tense is adopted more frequently, especially when referring to past events or figures.
For example:
‘In the republican times, the most well-known Shandong Laozi performer is GU 228 Hezhen 顧合真 who belongs to the Longmen School, and is mostly active in Fan county 229 範縣of He’nan province, and the rural areas of Heze 菏澤and Liaocheng 聊城, Shandong 230 province.’ (228-231)
If the term 'republican times' refers to the period between 1912-1949, it would be more appropriate to rewrite the paragraph using the past tense.
Again:
‘As for the Puyang Qinshu in the republican times, social investigations and statistics 288 are much more brief: Puyang Qinshu performers worship Quanzhen master QIU Chang-289 chun as patriarch, and belong to CHAI branch, one of the aforementioned four branches 290 affiliated with Longmen School.’ (288-291)
To avoid confusion, could you please clarify whether the statement above is referring to the present day or the historical period of the Republican era? It is unclear whether the worship of Master Qiu is being practised today or solely occurred during the Republican era. The use of present tense throughout the statement is contributing to this ambiguity.
2) The utilitarian argument that local performers utilised Quanzhen elements or fabricated their lineages purely to enhance their social standing is inadequately developed and demonstrates a deficiency in engaging with current scholarship on Daoist studies. Indeed, whilst the manuscript heavily relies on describing its primary sources, a closer engagement with the current historiography of Daoism would improve the arguments, making them more convincing (for specific examples, see my remarks below).
3) Likewise, the abstract states that ‘Quanzhen Taoism and its relationship with local performing arts is a very interesting subject in Quanzhen studies’. Interesting to whom? What is the relevance of this study? Why is it important? How does it posit itself in relation to the existing literature on Daoism and art/popular culture, local societies, etc.? Much has been written on all this, yet the author scarcely acknowledges or engages with previous scholarship. To be precise, it is not that the manuscript is irrelevant, but the author needs to provide a clear and explicit explanation to the readers as to why this manuscript is deserving of their attention. Merely stating that the subject is 'interesting' without additional elaboration and engagement with the relevant historiography is insufficient and does not make justice to its significance. For example, why is it important for us to gain a deeper understanding of the interplay between Daoism, Chinese performing arts and local societies? Or more specifically, why is it significant to know that Quanzhen Daoism influenced folk artists in Shandong and Henan?
As it currently stands, the paper is mostly descriptive rather than analytical. It would greatly benefit from having one or a few guiding questions that could be addressed throughout the manuscript, which would increase clarity and focus.
4) The ‘generation system’ of local performers and Quanzhen Daoism is a prominent topic in this manuscript, yet it is not clearly defined in the introduction or throughout the paper. It is important to elaborate on what this system entails and how it was formed, otherwise the reader cannot easily follow the arguments.
For example:
‘And moreover, living under the pressure of a hard life, local performers of different generations may become sworn brothers or get married, which also disrupts the regular successive sequence of the original Longmen lineage poem.’
In the absence of a clear explanation of the generation system, it becomes challenging for the reader unfamiliar with your sources to grasp the meaning of the original Longmen lineage poem and how the ‘pressure of a hard life’ might have disrupted it.
5) The beginning of the article provides an overview of the primary sources used in the research, but it lacks a discussion of the context in which these sources were produced. It is important to consider the background of those involved in producing the sources, as well as their motivations and biases. Were the sources created by government officials, scholars or local community members? Were they produced for a specific purpose, such as to document cultural practices for preservation or to promote a particular political agenda? By understanding the context in which the sources were produced, the reader can better evaluate the reliability and relevance of the information contained in them.
Furthermore, it would be valuable for the author to reflect on how the biases of the survey editors might have been shaped by the historical and cultural context in which they lived, and how their biases could have influenced the author's own statements. In the author’s own words, local performers ‘fabricated’, ‘deceived’, ‘distorted’. As explained below, I would adopt a more nuanced language. The author's use of such terminology suggests a suspicious, if not judgmental view of local performers, which is consistent with a particular historical tradition in China.
Specific comments:
‘But interestingly, even most local performers do not know more than their master, or grand master in some cases, they do know quite clearly that they are the successor de facto of the Taoist immortals.’ (327-329)
This statement looks interesting and could be developed further as a potential line of argument. It would be valuable to explore why it was crucial for those performers to emphasise their lineages as being from Daoist immortals and what sort of social implications it held. Additionally, it would be interesting to investigate whether this had an impact on their audience and how it may have influenced their perception of the performers. To effectively develop this line of argument, however, it is crucial to use both primary sources and appropriate secondary literature. Failing to do so may result in repeating the biases of the primary sources used, as demonstrated in the example below:
‘In fact, the reason why local performers are affiliated with Taoism is closely connected with their social status in the pre-modern society (L. Wang 1981; Me 2005). “In the feudal society, to avoid being insulted or humiliated by the society, performers claim that they are descending from the lineage of immortals; born with natural air, walking in the three worlds (Heaven, Earth and Human World) and equipped with five elements (wood, fire, earth,metal and water), they claim Three Pures (Sanqing 三清) and Five Patriarchs (Wuzu五祖) to be their ancestors.(QYHB, 2, p. 185)”’ (330-336)
First, the biases in this passage should be addressed in the section devoted to methodology and introduction to primary sources. This section may include information on the individuals or organisations responsible for collecting, recording and publishing folk customs during the periods of interest, as well as their potential biases and motivations. For example, is it just a coincidence that those scholars conducted their surveys in the 1930s, 1950s and 1980s?
Furthermore, even if the author agrees with the editors of the survey that local performers were ‘fabricating’ lineages to deceive their audiences, it is important to engage with the secondary literature to provide additional evidence and support for such a view. This can strengthen the argument and make it more convincing to readers who may have different perspectives or interpretations of the primary sources.
‘Meanwhile, the reason why local art performers choose Quanzhen Taoism to rely on is also because of its social status and propagating method […]’ (348-349)
The argument put forward in this paragraph lacks sufficient evidence to be convincing enough. The author claims that local performers appropriated Quanzhen Daoism solely for the purpose of raising their social status, citing the popularity of Quanzhen Daoism in the region as a reason for the adoption. However, this view requires closer engagement with the existing historiography. As it stands now, this claim sounds very much like the old anti-religion rhetoric championed by the Chinese state during the Republican and present-day Communist periods.
If the only motivation for the adoption of Quanzhen lineages is social legitimacy, then one would expect other professionals in Shandong and Henan, such as healers and martial artists, to have done the same. Without further analysis of these related fields, the argument lacks the necessary depth to support its claims.
Xu Liping is mentioned in passing in lines 63-64, and then return to him in lines 242-247. Who is Xu Liping? What is the importance of mentioning him? The way he is being presented in 242-247 looks quite random and disconnected.
The section ‘The Ever Spring Guild’ needs clarification. Firstly, the author stated that the Guild is (or should be ‘was’?) a guild of performers. However, the following paragraph listed other professions that were also part of the guild, causing confusion. Moreover, the purpose of this section needs to be clearly defined. If the author intends to use the Guild as an example of the Quanzhen connection with local performers, this argument should be made explicit earlier in the section. This will help the reader to better understand the significance of the Guild in the larger context of the manuscript.
Throughout the manuscript, it is taken for granted that certain elements are of Quanzhen origin, but the reasons that led to such a conclusion are seldom explicitly stated.
For example:
‘Based on the core factors 498 and procedure mentioned above, we could easily find that the performer’s ceremony 499 for acknowledging someone as the master is almost the same as the initiation and 500 ordination ceremony of the Quanzhen Longmen School.’ (498-501)
To facilitate the reader’s comprehension, the author should explain how a typical ordination ceremony of the Longmen School works.
Again:
‘The ceremony of new performer initiation has distinctive Quanzhen Taoist fea-427 tures. For example, when Shandong Kuaishu performer GAO Yuanjun formally 428 acknowledge QI Yongli as his master, he burns incense in front of the memorial tablet 429 “Heaven, earth, emperor, parents and master (Tian di jun qin shi 天地君親師)”, lo-430 cated in the middle of the hall; he then prostrates before his master, listening to his 431 master announcing: “I, a little Taoist priest in blue clothes, go down the hill under the 432 immortal master’s order; Today is no other than an auspicious day, and I will transmit 433 the Tao for the immortal master.”’
Why are these ‘distinctive Quanzhen Taoism features’? Does it mean that other Daoist lineages do not perform similar rituals? What makes this particular ritual Quanzhen in nature?
‘However, while they adopt the Quanzhen lineage poem and institutional form, the 584 local art performers greatly change the Quanzhen history, and rewrite the Longmen line-585 age poem with different characters. This may be due to the poor education of the perform-586 ers on the one hand, and their unfamiliarity with Quanzhen Taoism on the other hand.’ (584-587)
The passage above seems to contradict much of the manuscript, which states that local performers borrowed Quanzhen Daoism to raise their social status. But if many of them were unfamiliar with Quanzhen Daoism, how would this appropriation ever be possible? The author may need to revise their argument about the motivations behind the appropriation of Quanzhen Daoism.
‘Since all the local performing arts we are discussing here affiliate themselves with 595 the Longmen School, when tracing their own history, they always have to deal with the 596 origin and development of Quanzhen Taoism. But, most of their narratives are non-his-597 torical and unreliable, herein many Quanzhen immortals and masters are randomly re-598 placed by mythical and imaginary Taoist figures, or many non-Quanzhen immortals and 599 masters just appear in the Quanzhen history out of nowhere.’ (595-600)
How can we determine the reliability of their narratives? From whose perspective are they considered ‘unreliable’? With the limited evidence available – as stated in lines 58-80 – can we, today, truly conclude that the choices of local performers were random and came out of nowhere? Lack of historical evidence cannot be taken as proof that something did not exist or did not follow a particular logic. To offer a more nuanced interpretation of the sources in question, it would be beneficial to engage more closely with the latest historiography of Daoist studies produced within and outside Mainland China.
Again:
‘Based on the statements above, we could say that, in the process of imitating the Quanzhen institution, local performing arts made more or less deviation and variation, even fabrication in some aspects, which partly led to the inauthenticity and incredibility of their own history.’ (739-742)
Again:
‘As they have greatly changed the authentic history and then remade it according to their circumstances and needs, their orthodoxy is theoretically established.’ (647-649)
What does the author mean here by ‘authentic history’? Is it possible for any kind of history to be ‘authentic’ – as if objective?
‘For example, the phrase “希微衍自寧721 ”, which means “it is in what is held and subtle that one finds peace within oneself” 722 (Herrou, 2005, p.316), in the poem of Baiyun Monastery’s character is expressed as “西湖723 岩子寧” in Shandong Laozi and “喜為宴子林” in Guangzhou Dagu13, none of which can 724 be understood or explained by the local performers of these two genres. Facing such status 725 quo, we can say for sure that although their pronunciations are extremely close, the origi-726 nal doctrinal difference and philosophical meaning is getting lost.’ (721-727)
How is it possible to know the ‘original meaning’ of that passage? Original meaning according to the author of that passage? If so, who is the author, and how do we know that the translation provided by Herrou corresponds to the meanings originally intended by the author? Questions of ‘deviation of meaning’ are extremely difficult to prove in historical terms. An important theoretical reference on this matter is Forms and Meanings: Texts, Performances, and Audiences from Codex to Computer, by Roger Chartier.
‘On some level and to some scholars of Quanzhen Taoism, these variations or devia-650 tions from the standard history of Quanzhen Taoist succession may seem random or ca-651 pricious.’ (650-652)
What scholars? The entire paragraph lacks references.
‘Scholars of Taoism all know for sure that Hongjun laozu, 611 ZHANG Guolao, XU Maogong and WANG Chongyang are either mythical or real Taoist 612 figures, however, their successive relations from master to disciple in Guangzhou Dagu 613 are far from standard in the various hagiographical accounts.’ (611-614)
This statement needs appropriate referencing. ‘Either mythical or real Taoists’ sounds like a tautology (what else could they be?). The author's claim that a narrative not following the ‘standard hagiographical account’ must be fabricated or false should be supported by evidence or sources from the relevant historiography in Daoist studies. The author could also explain why they consider written records to be more reliable or trustworthy than oral accounts, and engage with any relevant scholarly debates on this issue. Ultimately, it would be helpful for the author to provide a more nuanced argument that considers different perspectives and interpretations within the field of Daoist studies.
‘seemingly alike but specious’ . The author uses this sentence quite a few times, but it is unclear what it means in practice within the context of this manuscript.
‘Institutional charisma’ is an important concept in the manuscript and religious studies. It would be important to define it earlier in the text, and return to it in the conclusion, linking it to the discussion on local performers in Shandong and Henan.
Suggestions on spelling:
Taoism is an old spelling, whilst Daoism is the standard spelling nowadays. Any special reason not to adopt Daoism?
Why He’nan and not the pinyin Henan?
If there is a reason for Taoism and He’nan, please clarify your choice in a footnote.
Chongyang pai” 重陽牌 (625): 牌 or 派?
